# Mechanisms of Hormonal, Genetic, and Temperature Regulation of Germ Cell Proliferation, Differentiation, and Death During Spermatogenesis

**DOI:** 10.3390/biom15040500

**Published:** 2025-03-29

**Authors:** María Maroto, Sara N. Torvisco, Cristina García-Merino, Raúl Fernández-González, Eva Pericuesta

**Affiliations:** 1National Institute for Agricultural and Food Research and Technology (INIA-CSIC), 28040 Madrid, Spain; maria.maroto@inia.csic.es (M.M.); cristinagmerino@gmail.com (C.G.-M.);; 2School of Agriculture and Food Science, University College Dublin, D04 W6F6 Dublin, Ireland; sntorvisco@gmail.com

**Keywords:** spermatogenesis, hormone, heat stress, gene regulation, cellular mechanisms

## Abstract

Spermatogenesis is a complex and highly regulated process involving the proliferation, differentiation, and apoptosis of germ cells. This process is controlled by various hormonal, genetic, and environmental factors, including temperature. In hormonal regulation, follicle-stimulating hormone (FSH), luteinizing hormone (LH), and testosterone (T) are essential for correct spermatogenesis development from the early stages and spermatogonia proliferation to germ cell maturation. Other hormones, like inhibin and activin, finely participate tuning the process of spermatogenesis. Genetic regulation involves various transcription factors, such as *SOX9*, *SRY*, and *DMRT1*, which are crucial for the development and maintenance of the testis and germ cells. MicroRNAs (miRNAs) play a significant role by regulating gene expression post-transcriptionally. Epigenetic modifications, including DNA methylation, histone modifications, and chromatin remodelling, are also vital. Temperature regulation is another critical aspect, with the testicular temperature maintained around 2–4 °C below body temperature, essential for efficient spermatogenesis. Heat shock proteins (HSPs) protect germ cells from heat-induced damage by acting as molecular chaperones, ensuring proper protein folding and preventing the aggregation of misfolded proteins during thermal stress. Elevated testicular temperature can impair spermatogenesis, increasing germ cell apoptosis and inducing oxidative stress, DNA damage, and the disruption of the blood–testis barrier, leading to germ cell death and impaired differentiation. The cellular mechanisms of germ cell proliferation, differentiation, and death include the mitotic divisions of spermatogonia to maintain the germ cell pool and produce spermatocytes. Spermatocytes undergo meiosis to produce haploid spermatids, which then differentiate into mature spermatozoa. Apoptosis, or programmed cell death, ensures the removal of defective germ cells and regulates the germ cell population. Hormonal imbalance, genetic defects, and environmental stress can trigger apoptosis during spermatogenesis. Understanding these mechanisms is crucial for addressing male infertility and developing therapeutic interventions. Advances in molecular biology and genetics continue to uncover the intricate details of how spermatogenesis is regulated at multiple levels, providing new insights and potential targets for treatment.

## 1. Introduction

Spermatogenesis is a complex process that takes place within the seminiferous tubules of the testes, by which haploid male gametes, known as spermatozoa, are produced in mammals [1]. This process requires a highly specialized environment to support the development and maturation of sperm cells that is provided by somatic cells like Sertoli, Leydig, and peritubular myoid cells (PTM). Nurtured by hormones and growth factors secreted to the environment, germ cells perform proliferation, differentiation, and spermiogenesis, involving spermatogonia, spermatocyte, and spermatid to produce mature sperm [2]. The proliferation phase, known as the mitotic phase, involves the mitotic division of spermatogonia stem cells (SSC) located along the basal membrane of the seminiferous tubules. These rounds of mitotic divisions drive the expansion of the spermatogonial population before entering the differentiation stage, ensuring a continuous supply of germ cells throughout the reproductive life of the male. During differentiation, the primary spermatocytes transition into haploid cells. The process begins with meiosis I. Each secondary spermatocyte then undergoes meiosis II, resulting in four haploid spermatids (Figure 1) [1]. Throughout meiosis, chromosomal recombination and crossover events ensure genetic diversity among the sperm cells. The final stage, spermiogenesis, involves the morphological and biochemical transformation of round spermatids into elongated spermatozoa. This includes the condensation and compaction of DNA to facilitate motility, the formation of the acrosome (an enzyme-containing cap essential for fertilization), and the development of a flagellum for movement. DNA compaction is achieved through the replacement of histones with protamines, resulting in a highly compact and transcriptionally inactive chromatin structure [3]. Along the whole process, apoptosis results in a crucial regulatory mechanism that eliminates defective or excess germ cells. It ensures that only genetically intact and properly differentiated sperm are produced, maintaining the balance between germ cell proliferation and maturation within the seminiferous tubules [4].

In this review, we provide an updated perspective on the critical role of hormonal, genetic, and temperature regulation during spermatogenesis, and how they participate in the cellular mechanisms of germ cells through the process.

## 2. Hormonal Regulation in Spermatogenesis

Hormones play a crucial role in spermatogenesis by coordinating essential functions and interactions through the hypothalamic–pituitary–gonadal axis (HPG) (Figure 2). This axis is of vital importance in gametogenesis regulation in vertebrates, both for spermatogenesis in males and folliculogenesis in females [6]. Some studies have elucidated the evolutionary origin of the HPG axis, stating sea lampreys as the earliest evolved vertebrates. In addition, some functional roles have been attributed to gonadotropin releasing hormone (GnRH), which acts via the HPG axis regulating reproductive mechanisms [7,8]. This hormone, GnRH, is the key regulator of the HPG axis, which is secreted by the hypothalamus in pulses. Its function lies in stimulating the anterior pituitary gland (AP) to synthesize gonadotropins, such as follicle-stimulating hormone (FSH) and luteinizing hormone (LH), and its subsequent secretion by gonadotropic cells. To reach their site of action, gonadotropins enter to the circulatory system, and they are transported to the testis, where FSH and LH act through specific G protein-coupled membrane receptors (GPCR). FSH targets Sertoli cells, which, together with the developing sperm cells, comprises the cellular component of the seminiferous tubules, whose function is to support spermatogenesis, facilitating the progression of germ cells and to nurture them in promoting the development of spermatocytes and spermatids. In addition, Sertoli cells secrete Inhibin B, which helps to regulate FSH release by negative feedback on AP, and anti-Müllerian hormone (AMH), which is responsible for the female reproductive structures’ regression during male development.

Moreover, FSH stimulation, via FSH receptor (FSH-R) binding in Sertoli cells, activates the canonical cAMP/PKA pathway, leading to the secretion of various growth factors, such as stem cell factor (SCF) and c-kit ligand that support the survival, proliferation, and differentiation of germ cells. Nonetheless, it is now clear that this is just one of several mechanisms activated by FSH stimulus to transduce intracellular signals [10]. Thus, FSH plays an important role, both independently or in combination with testosterone (T), in the proliferation of Sertoli cells, as well as in supporting spermatid maturation by producing signalling molecules and nutrients for this matter. After puberty onset, when sexual maturity has been reached, FSH, together with T, triggers signals mediated by Sertoli cells to propagate germ cell maturation, supply antiapoptotic survival factors, and regulate adhesion complexes between germ and Sertoli cells [11]. The effects of FSH have been classically studied using hypophysectomised or GnRH immunized animal models treated with exogenous FSH [12] or human patients with hypogonadotrophic hypogonadism [13]. These studies have shown that FSH enhances germ cell proliferation by increasing spermatogonia and spermatocytes count, but it is unable to produce spermatids in the absence of T. Additionally, results from studies involving FSH-R knockout animal models [14,15] have suggested that FSH is not essential to maintain fertility, in line with that observed in studies with hypogonadotrophic patients [16].

FSH is also necessary for spermatogonia to differentiate into primary spermatocytes. It regulates the meiotic entry by inhibiting Activin A (which promotes DNA synthesis in spermatocytes) and activating its counteracting factors Inhibin B and IL-6 (via PKC dependent pathway). Also, it has been described in murine Sertoli cells that FSH promotes nociception expression via cAMP/PKA. Additionally, in order to promote spermatocyte survival, FSH signalling was described inhibiting both intrinsic and extrinsic apoptotic pathways through the expression of Galectin-3 [17,18].

On the other hand, LH influences testicular interstitial Leydig cells. These cells are responsible for androgens production, as well as for insulin-like peptide 3 (INSL3) [6,17]. LH’s main function is to stimulate Leydig cells to produce T, which is one of the main testicular androgens, playing a crucial role in the regulation and progression of spermatogenesis in mammals [19]. Concretely, it is essential for the successful development of male secondary sexual characteristics. These two hormones act together as a LH/T signal and play a critical role in initiating and maintaining spermatogenesis. In Figure 3, we summarize the classical and non-classical pathways in Sertoli cells.

T and its metabolite dihydrotestosterone (DHT) bind to androgen receptors (AR) in Sertoli cells, diffusing through the plasma membrane and forming a complex that activates the classical signalling pathway (Figure 3). Then, AR dimers are translocated to the nucleus, targeting a broad spectrum of genes that can promote Sertoli cells maturation, induce SSC differentiation, ensure spermatocyte meiosis, and safeguard blood–testis–barrier (BTB) integrity. Conversely, the non-classical signalling pathway starts with translocation of the AR from the cytoplasm to the plasma membrane, where it interacts with SRC proto-oncogene (*Src*), causing epidermal growth factor receptor (EGFR) phosphorylation, which activates MAPK kinases to regulate targeted gene transcription. This pathway, as well as the classical one, is involved in BTB integrity and spermatocyte meiosis, as well as in the adhesion of Sertoli cells to spermatids, spermatid development, and sperm release [17,20]. Other non-classical pathways stimulated by T have been reviewed in J. M. Wang et al., 2022 [20], as well as, upstream, downstream, and transcription regulators participating in this complex network triggered by AR signalling during spermatogenesis. In contrast to the observations made in FSH-R knockout mouse, which was fertile, the deletion of LH receptor (LH-R) in male mice has been reported to greatly reduce T levels in serum and testis and to impair spermatogenesis completion [12]. The explanation for this relies on the activation of adenylate cyclase and the increase in cyclic AMP (cAMP) levels after LH-LHR binding on the Leydig cells’ membrane, which in turn, activates protein kinase A (PKA), activating the steroidogenic acute regulatory protein (StAR), finally leading to T synthesis from cholesterol. A study in male rats exposed to 17α-ethynylestradiol (EE2), an endocrine-disrupting chemical, has reported that EE2 treatment reduces T secretion by the downregulation of the LH-R pathway, leading to a decrease in cAMP, which in turn, suppresses steroidogenesis by the downregulation of some steroidogenic enzymes, such as P450 side-chain cleavage enzyme (P450scc) and StAR. These results propose a model to further investigate the mechanisms underlying T inhibition [21].

It was reviewed in A. Kumar et al. (2018) [22] that spermiation might be controlled by FSH acting via androgen. In one of the studies, in the model of suppression of FSH and androgen, the transcription of genes that participated in lysosome function and lipid metabolism in Sertoli cells was disturbed [23]. Also, other hormones have been described to play a role in spermiation, such as oestrogen contributing to actin and cytoskeletal dynamics, retinoic acid (RA) for cytoskeletal remodelling, oxytocin for tubular contractibility, among others. Moreover, this step has been considered a proper target for male contraception, since it is a delicate phase in different species, such as rodents, monkeys, and humans [22].

Other key hormones that are worth mentioning on this complex regulatory network are prolactin (PRL) and oestradiol (E2). PRL is synthesized and secreted by the AP and binds to its receptor PRLR on several tissues, including testis, epididymis, seminal vesicle, and prostate, activating the JAK-STAT signalling pathway and playing a role in male gonadal function by regulating the number of LH and FSH receptors, releasing gonadotropins, and promoting steroidogenesis and spermatocyte–spermatid conversion, among other effects [24,25]. However, some KO studies for PRL or PRLR have attempted to further investigate their effect in male reproductive functions, but the results were not conclusive, leaving the role of this hormone and its receptor unclear in relation to male fertility [26,27]. Regarding E2, it is produced in the testis in small amounts by the aromatization of T, by action of the enzyme CYP19A1 or aromatase. E2 acts by binding to oestrogen receptors ERα and Erβ; although, only Erα has been reported to have a role in maintaining male fertility [28]. Additionally, the disruption of *Cyp19* gene coding for aromatase (ArKO mice) results in decreased fertility in males of advancing age [29].

## 3. Genetic Regulation in Spermatogenesis

The genetic mechanisms involved in the complex process of spermatogenesis guarantee precise cell differentiation and stage-specific gene expression due to three different regulatory levels: gene regulation, transcriptional regulation, and epigenetic regulation [30]. While numerous genes, transcription factors, and other key elements are involved in spermatogenesis regulation, this review focuses on the most relevant and/or extensively studied ones, as shown in Table 1. Key transcription factors involved in the genetic regulation of spermatogenesis include *SRY* (sex-determining region Y) and *SOX9* (SRY-box transcription factor 9), both essential for sex determination and the development of male germ cells. *Sry*, located at the distal region of the short arm of the Y chromosome, acts as the master regulator, initiating testicular differentiation by activating the expression of *Sox9* in Sertoli cells [31]. In turn, *Sox9* promotes the transcription of genes involved in the formation and maintenance of seminiferous tubules, such as AMH (anti-Mullerian hormone), and in the regulation of the environment necessary for spermatogenesis [32]. *Plzf* (ZBTB16) (promyelocytic leukaemia zinc finger) is also a key transcription factor expressed in undifferentiated spermatogonia and is critical for SSC maintenance, where *Plzf* mutation results in a progressive germ cell loss [33]. This factor regulates SSCs by direct promoter binding to differentiation-related genes, such as *Sohlh2*, *Kit*, *Stra8,* and *Dmrt1*, repressing them [34]. *Plzf* knockout models are infertile and present testicular degeneration [35]. *SOHLH2*, expressed in type A spermatogonia, regulates spermatogenesis by binding and modulating genes, such as *Gfra1*, *Ngn3*, and *Sox3* [36]. Additionally, *SOHLH2* promotes *Kit* expression, while its downregulation results in increased PLZF levels and reduced *Kit* expression [36,37]. Another key regulator, *Stra8*, an essential target of RA, governs spermatogonial differentiation; its deficiency in mice leads to spermatocyte depletion and disrupts meiotic progression [38]. Furthermore, *Dmrt1* (doublesex–mab3-related transcription factor gene family) is a differentiating marker [39] and a sexual determination gene [34]. It acts by limiting RA response on spermatogonia, downregulating *Stra8* (Stimulated by RA 8) transcription. *Dmrt1* also activates the spermatogonial differentiation factor *Sohlh1* (spermatogenesis- and oogenesis-specific basic helix-loop-helix 1) transcription, thereby blocking meiosis and inducing spermatogonia development [40]. In addition, there are other genes that play a crucial role in human spermatogenesis and have significant clinical relevance. The AZF (azoospermia factor) region on the Y chromosome comprises critical genes whose microdeletions have a direct impact on spermatogenesis, such as RNA-binding motif protein Y-linked family 1, *Ptpn13* Like Y-linked, deleted in azoospermia and DEAD-box Y-linked (*Rbmy1*, *Pry*, *Daz*, and *Dby*), are directly involved in essential germline-specific processes and hold potential for improving future therapeutic strategies [31]. DAZL (deleted in azoospermia-like) belongs to the DAZ family of RNA-binding proteins. *Dazl* is expressed during embryonic and adult gametogenesis, playing a crucial role in germ cell determination, development, and meiotic progression [41,42]. DAZL deficiency impairs the proliferation and differentiation of spermatogonial progenitors as a result of translational regulation [40], and *Dazl* knockout results in infertility in both sexes [43]. Recent findings by Mikedis et al. (2020) [44] establish how *Dazl* drives the expansion and differentiation of spermatogonial progenitors. Furthermore, DAZL enhances germ cell development, promoting the translation of genes that regulate spermatogonial proliferation and differentiation [44]. Moreover, studies have found different transcription factors expressed in a stage-specific manner during spermatogenesis, such as *Tbx3* (T-box transcription factor 3) [45], activated in infant spermatogonia [46], and *Utf1* (undifferentiated embryonic cell transcription factor 1) in puberty [39]. On the other hand, some transcription factors, such as cAMP response element modulator (CREM) and testis-specific CREM (CREMt), act through a testis-specific isoform generated by alternative splicing, controlling post-meiotic germ cell differentiation [30]. Its absence generates a knock-on effect that deactivates a large number of genes specific for sperm differentiation, triggering male infertility [32].

Post-transcriptional gene control regulation takes an essential role, especially in the later steps of sperm differentiation when chromatin compaction induces transcriptional silencing in elongating spermatids [47]. NANOS proteins are highly conserved post-transcriptional regulators. Studies involving knockout models have highlighted the critical role of *Nanos* genes in germ cell development [48]. In mice, these proteins interact with the CCR4-NOT deadenylation complex to mediate the downregulation of target mRNAs [49]. The absence of NANOS2 and NANOS3 leads to a progressive depletion of primordial germ cells (PGCs) [50], a phenotype also observed in other RNA-binding protein (RBP) knockout models, such as TIAL1 and DND1 [51,52]. RBM46 is a novel RBP essential in spermatogenesis. Similar to other RBPs, RBM46 regulates mRNA translation and stability (Y. Lv et al., 2023). *Rbm46* knockout models exhibit testicular atrophy, reduced testis weight, and infertility, with seminiferous tubules lacking spermatids [53,54]. RBM46 is crucial for meiosis, as its absence leads to spermatocytes arresting in the early meiotic stages due to defects in chromosome synapsis and premature entry into the aberrant metaphase [53,54]. Additionally, RBM46 targets genes involved in chromosome segregation and the mitotic-to-meiotic transition, coordinating with RNA-processing cofactors, such as YTHDC2/MEIOC, to ensure proper meiotic entry [54]. Due to the high requirements for the control of gene expression, it is not surprising that microRNAs have provided an important additional level to the regulation of sperm production. MiRNAs play a crucial role in spermatogenesis and testicular function, regulating the proliferation and differentiation of both Sertoli and germ cells [55]. It has been studied how disrupting the enzymes and proteins involved in miRNA biosynthesis leads to fertility disorders in males, such as DICER, whose absence in mice spermatogonia causes increased infertility due to several deficiencies, including defects in spermatogonial progression and abnormalities in sperm elongation, apoptosis in Sertoli cells, impaired spermatogenesis and sperm production, and azoospermia, among others [56,57,58]. Moreover, DROSHA suppression in spermatogenic cells leads to severe germ cell depletion and low counts of elongating spermatids [59]. 

The balance between self-renewal and the differentiation of SSCs is vital for spermatogenesis, with miRNAs regulating key factors, like glial cell line-derived neurotrophic factor, fibroblast growth factor, B cell CLL/lymphoma 6 member, and Ets transcription variant 5 (*Gdnf*, *Fgf*, *Bclb6*, and *Etv5*), which support SSC self-renewal and maintain the SSC niche. Additionally, miRNAs have an influence on proteins such as PLZF and RA, which promote spermatogonial differentiation [60]. Among the critical miRNAs involved in spermatogenesis regulation, miR-146 overexpression inhibits RA-mediated spermatogonial differentiation by targeting *Med1*, a coregulator of RA receptors, and decreasing *Kit* proto-oncogene expression [61]. Similar to *Dmrt1*, *Kit* is also a differentiating marker [39]. miR-221/222 maintain the undifferentiated state of spermatogonia by repressing *Kit* expression, thus preventing their transition to differentiated spermatogonia [62]. Another crucial miRNA is miR-202-3p, regulated by GDNF and RA, and its knockout mice lead to premature SSC differentiation, highlighting its importance in SSC maintenance [63]. These miRNAs are key to regulating the balance between SSC self-renewal and differentiation.

Male germ cells exhibit a highly diverse transcriptome, particularly in round spermatids and pachytene spermatocytes, driven by extensive transcriptional activity involving both protein-coding and long non-coding RNA (lncRNAs) genes [64]. In spermatogenesis and testicular development, some of the key studied lncRNA-coding genes are *Mrhl*, which has been shown to be critical for regulating Wnt signalling, and plays a critical role in spermatogonia development [65]; *Tsx* (a testis-specific X-linked gene), which is specifically expressed in pachytene spermatocytes (*Tsx* knockout mice exhibited decreased testicular weight and increased spermatocyte apoptosis) [66]; and *Dmr* (*Dmrt1*-related gene), which plays a negative regulatory role for Dmrt1 in male sexual development [44]. Numerous studies have established that several lncRNAs serve as a fundamental element in regulating SSCs, maintaining SSC survival, and self-renewal through protein-coding genes and miRNA [67]. Functional studies, as carried out by Weng et al. (2017) [68], found specific lncRNAs directly involved in the regulation of key genes essential for spermatogenesis and testis development, such as LNC_003902 targeting *GATA4* (transcriptional regulator for development of the murine foetal testis) and LNC_007258 targeting *NKAP* (involved in maintenance and differentiation of spermatogonial stem cells in spermatogenesis). As shown in the study by Liang et al. (2019) [69], there is a tight correlation between lncRNA Gm2044 and miR-202 in spermatogenesis, where both RNAs are upregulated in cases of non-obstructive azoospermia (NOA), a condition characterized by spermatogonial arrest. The identified lncRNA Gm2044 plays a significant role in the context of male infertility as a miR-202 host gene, directly influencing the expression of the miRNA target gene, *Rbfox2*, which is crucial for regulating cell proliferation and differentiation in spermatogenesis. Another important way for lncRNAs to participate in epigenetic regulation during spermatogenesis is recruiting histone modification-related enzymes, such as histone acetylation and methylation, establishing specific gene expression patterns for different stages of development [70]. Moreover, they can regulate chromatin activity by recruiting histone modification enzymes [71]. 

In addition, Piwi-interacting small RNAs (piRNAs) also play a central role in spermatogenesis. PiRNAs silence transposable elements (TEs) during meiosis, protecting the integrity of the germline genome [47,72]. 

Similarly, circular RNAs (circRNAs) are gaining attention due to their ability to act as miRNA sponges, inhibiting their function and then regulating gene expression networks [73]. Although their specific function in spermatogenesis is less explored, new studies suggest their involvement in NOA [74]. By inhibiting miRNAs, protein binding regulation and gene transcription regulation can lead to cell cycle arrest and spermatogonia apoptosis, affecting spermatogenesis [73]. 

In order to preserve genome integrity and also promote genetic diversity, germ cells undergo epigenetic reprogramming during development and are conditioned to attain full developmental potential upon fertilization [75]. 

It is known that epigenetic regulation is indispensable, not only in general gene expression but also in the precise control of spermatogenesis, as it affects key processes, such as germ cell differentiation and fertility. N6-methyladenosine (m6A) is one of the most widespread epigenetic modifications in mRNA and plays a crucial role in gametogenesis across multiple species. As is demonstrated in Fang et al. [76], disruptions in m6A regulation or abnormal levels of this modification are closely linked to defects in gametogenesis and embryonic development. Genome-wide methylation studies revealed how the sperm epigenome significantly differs from somatic cells due to unique DNA methylation patterns and the presence of specialized chromatin structures required for meiosis and spermiogenesis. These differences highlight the distinct roles that DNA methylation in sperm plays beyond gene expression, influencing chromosomal organization and fertility-related processes [77]. The *Dnmt* family, including *Dnmt1*, *Dnmt3a*, *Dnmt3b*, and *Dnmt3l*, encodes enzymes that regulate DNA methylation, adding methyl groups to cytosines in CpG contexts, essential for imprinting and germline-specific methylation. In male germ cells, *Dnmt3a* and *Dnmt3l* establish de novo methylation during prenatal development, and disruptions can lead to infertility due to defective imprinting and meiotic failure [78,79], while *Dnmt1* maintains these patterns postnatally [80]. 

Chromatin condensates by the significant remodelling process called protamination. This crucial step replaces histones with protamines and involves different modifications and specific proteins. Aside from DNA compaction, protamination is also essential in epigenetic regulation, because disrupting the role of histones in shaping the embryonic epigenetic landscape can lead to changes in gene expression and impaired fertility. This emphasizes their crucial function in epigenomic regulation during spermatogenesis [81]. Due to their importance, the regulation of protamines *Prm1* and *Prm2* transcription, translation, and final gene expression is carefully modulated [82,83]. Among key regulators, previously mentioned CREM transcription factor, *Taf7l*, and *Tarbp2* also take an important role in the regulation, as it has been previously proved how their absence led to protamines failed translation, reduced sperm count, and motility and male infertility [81,84,85]. *Sycp3* has also been widely studied, as it has a major role in meiotic chromosome compactation, acting as a structural scaffold that facilitates synapsis between homologous chromosomes [86]. Knockout mice show a sexually dysmorphic phenotype and infertility due to cell death in early meiosis [87]. New in vitro analysis suggests that meiotic chromosome compaction is driven by SYCP3’s DNA-binding and self-assembly, while in vivo, its function relies on SYCP2 and the meiotic cohesin core [86]. 

These principal elements described above are summarized in Table 1 and represented in the gene map in Figure 4.

## 4. Regulation by Temperature

Mammalian body temperature is regulated by various mechanisms that balance heat production and temperature loss. Internal organs are the primary heat generators due to their metabolic activities, which increase intra-abdominal temperature. The testes, in particular, generate significant heat during spermatogenesis. They must mitigate this heat as an increase of just 1.5–2 °C in testicular temperature can inhibit spermatogenesis [88]. Moreover, normal testicular function requires a substantial temperature reduction of 2–8 °C below core body temperature [89].The localization of male reproductive organs is one of nature’s strategies to prevent heat damage. The suspension of the testes outside the abdomen, in the scrotum, is a distinctive feature of homeothermic animals, except for marine mammals and elephants. The presence of the scrotum establishes an abdominal–scrotal–testicular temperature gradient that maintains the testes within a functional temperature range lower than the abdominal core [90]. In addition, the following mechanisms contribute to the testicular thermoregulatory system: the pampiniform plexus surrounding the testicular artery; the tunica dartos muscle, a thin rugated fascial muscle of the scrotum that reduces scrotal volume and brings the testes closer to the warmer abdominal region; and the cremaster muscle, a thin fascial muscle of the spermatic cord that retracts the testes. In contrast, cetaceans, which have intra-abdominal testes, rely on a counter-current heat exchanger (CCHE) instead of the pampiniform plexus [91]. The CCHE is formed by a spermatic arterial plexus juxtaposed to the lumbo-caudal venous plexus, which carries blood from superficial veins of the dorsal fin and flukes cooled by external water. This mechanism cools the spermatic arterial plexus that supplies the testes [92,93]. Cryptorchidism, a condition where the testes fail to descend into the scrotum, exemplifies the detrimental effects of elevated testicular temperatures. Studies have demonstrated the negative impact of increased temperature on spermatogenesis in animals with undescended testes or thermal stress induced in the laboratory [94]. High temperatures impair regulatory mechanisms, leading to the production of reactive oxygen species (ROS) [95]. This results in oxidative stress in germ cells, increased DNA damage, apoptosis, and autophagy [96,97], as well as alterations in gene expression. These changes reduce testicular function and sperm quality, contributing to male infertility [98,99]. 

Increased apoptosis due to heat stress leads to decreased testicular weight, germ cell loss, and low sperm concentration [100,101,102,103]. Even worse, after heat stress, a high amount of germ cells can carry out the complete development into abnormal morphological spermatozoa with damaged DNA avoiding apoptosis, which leads to a reduction in viability or motility [104,105,106]. In IVF experiments, heat stress has been linked to reduced sperm binding to the zona pellucida and DNA fragmentation, observed by significant declines in COMET assay parameters, such as tail length and tail moment [107]. Mitochondrial dysfunction caused by heat stress, including reduced ATP synthesis, decreased membrane potential, and lower mitochondrial protein content, further impairs sperm motility [108]. As heat stress increases oxidative stress, this affects Na^+^/K^+^-ATPase activity that reduces sperm flagellum movement, which explains abnormal motility and fertility reduction [109]. The phosphorylation state of glycogen synthase kinase 3α (GSK-3α), mediated by Ser21 phosphorylation, is associated with sperm motility in various species, including humans [110], bovines [111], and pigs [108]. At this point, Gong et al. (2017) [108] demonstrated that heat stress reduced levels of GSK-3α phosphorylation in board sperm, affecting its motility.

Heat stress also affects DNA integrity and gene expression. The synaptonemal complex, essential for homologous chromosome pairing during meiosis, is disrupted by elevated temperatures, leading to recombination failures that become DNA fragmentation and unpaired chromosomes or aneuploidy, damaging DNA integrity [94,112]. On one hand, heat stress induces mRNA and protein degradation and downregulates genes related to meiosis and DNA synthesis [113]. On the other hand, heat stress activates some genes related to defence, regulation, and DNA damage repair, as well as heat shock proteins (HSPs). Heat shock proteins are essential for spermatogenesis to protect proteins not only against heat but radiation and chemical exposure. HSP70 are highly conserved proteins across phyla, from archaebacterial to plants and animals. There are two members of the HSP70 family expressed specifically during spermatogenesis HSP70-2 and HSC70t. HSP70-2 protein is present in spermatocyte during the meiotic phase of spermatogenesis [114], while HSC70t is synthesized in post-meiotic spermatids [115]. HSP70-2 is associated with synaptonemal complex, and it is necessary for the formation of Cdk1cyclin B1 heterodimer. Male mice deficient in *hsp70-2* are infertile due to arrested germ cell meiosis in prophase I and increased apoptosis in the spermatocyte stage.

However, the effect of increased temperature in the testicles not only affects spermatogenesis and male fertility. Moreover, changes in gamete during spermatogenesis caused by heat stress do not compromise viability, but biology may also affect embryo development. Different studies demonstrated a reduction in the cleavage rate [107], as well as the preimplantational development of embryos produced in vivo [116] and in vitro [116,117,118]. Male mice subjected to heat stress sire fewer post-implantation embryos and foetuses with reduced body weights, not affecting resorptions [107,117,118,119]. In sheep, heat stress increases the proportion of degenerated embryos [120]. Paternal heat stress also alters protein profiles in preimplantation embryos, indicating its impact on embryonic development [121]. Additionally, scrotal heat stress can distort offspring sex ratios in mice, when males are mated with females on the same day as scrotal heat treatment, reducing the proportion of male offspring without affecting the ratio of X- and Y-chromosome-bearing spermatozoa [94]; although, it has been seen that the altered temperature environment could differently affect the functionality of spermatozoa carrying X or Y chromosomes for a short period, more than the spermatogenesis itself [94]. 

In humans, the most common origin of heat stress is the modern lifestyle that impairs proper testicular thermoregulation, such as the regular use of tight clothes that reduces airflow to the genital area [122,123,124], sedentary behaviour or seated position for long periods [125,126], obesity [127], or the use of thermal treatments [128,129]. 

In mammals, pathological heat stress is produced by cryptorchidism [130], varicocele when abnormal dilatation of the veins of the pampiniform plexus take place [131], or febrile episodes [132]. 

Additionally, global warming is a matter of concern that affects both animals and human fertility, contributing to its decline. Not only is the increase in the environmental temperature since the beginning of the industrial era well-documented but so is the increase in the frequencies of heatwaves and increased mortality rate due to heatstroke, dehydration, or hyperthermia. Different research has studied how the climate change affects fertility through the association between the season environmental temperature and sperm parameters, highlighting a likely detrimental effect of extreme temperatures [133,134,135,136]. 

## 5. Cellular Mechanisms of Proliferation and Differentiation of Germ Cells

In mammals, the formation of testes begins with the segregation of seminiferous cord bundles from the surface epithelium, which are subsequently divided into individual cords by mesenchyme (Figure 5A). At this stage, the primordial germ cells (PGCs) are named gonocytes, and a few days after birth, they undergo a transformation into SSCs. In the testis of adult mice, SSCs are identified as As spermatogonia (SA in Figure 1), where “s” stands for single, reflecting their undivided state and lack of intercellular bridges. Whether this dual pathway from gonocytes to spermatogonia is mouse-specific or a general mechanism from mammals remains uncertain; although, considering the parallel with Drosophila, it could be broadly used to the direct development from PGCs to spermatogonia [137]. 

The “seminiferous epithelium cycle” was defined in rats by Leblond and Clermont in 1952 by considering the division of the epithelium into separate stages, according to the cellular associations observed in each tubular cross-section. In rats, the seminiferous epithelium cycle was divided into 14 stages (I to XIV), and spermiogenesis was split into an additional 19 steps (1 to 19) [141,142]. In mice, the cycle is divided into 16 stages (I to XVI), as is showed in Figure 5C. The seminiferous epithelium cycle and spermatogenesis presents a different duration in each species (Table 2).

This spermatogenesis occurs within the seminiferous tubules, the functional unit of the mammalian testis, being regulated by several endocrine factors, such as testosterone, FSH, LH, and oestrogens. These seminiferous tubules are composed by Sertoli and germ cells [154]. It begins at puberty after a long preparatory period, called prespermatogenesis, present in the foetus and in the infant (Figure 5B) [155]. 

### 5.1. Phases of Spermatogenesis

Mammalian spermatogenesis is divided into the following three general phases: the mitotic division of SSC, the meiosis of the spermatocyte, and spermiogenesis. The proliferative phase, known as the Mitotic phase, involves the mitotic division of SSC located along the basal membrane of the seminiferous tubules. These rounds of mitotic divisions produce primary spermatocytes driving to the expansion of the spermatogonial population before entering the differentiation stage, ensuring a continuous supply of germ cells throughout the reproductive life of the male. During differentiation, the primary spermatocytes undergo meiosis to reduce their chromosome number by half, transitioning into haploid cells. The process begins with meiosis I, where a primary spermatocyte (diploid, 2n) divides to form two secondary spermatocytes (haploid, n). Each secondary spermatocyte then undergoes meiosis II, resulting in four haploid spermatids. Throughout meiosis, chromosomal recombination and crossover events ensure genetic diversity among the sperm cells. The final stage, spermiogenesis, involves the morphological and biochemical transformation of round spermatids into elongated spermatozoa (Figure 1).

#### 5.1.1. Mitotic Phase

In the proliferative stage or spermatocytogenesis, the SSC located adjacent to the basement membrane produces type A and B differentiated spermatogonia. Therefore, undifferentiated type A spermatogonia are amplified and self-renewed by mitosis to replenish the stem cells. A series of mitotic cell divisions with incomplete cytokinesis (since daughter cells stay interconnected by cytoplasmic bridges) first produce a pair of spermatogonial cells and then differentiate into 4–16 or even 32 aligned A spermatogonias, which migrate from the base compartment to the luminal compartment of the seminiferous tubules. Aligned A spermatogonia transition is regulated by at least one extrinsic factor (retinoic acid) and multiple intrinsic factors. A1 spermatogonia undergo subsequently six mitoses, generating A2, A3, A4, intermediate (In), and B spermatogonia, all of them being differentiated spermatogonia. The mitosis in mammalian gametogenesis is controlled by the following several components: the oscillation activity of cyclin/CDKs, the activity of anaphase-promoting complex/cyclosome (APC/C), the components at the mitotic spindle, such as small kinetochore-associated protein (SKAP), DNA repair-associated genes, several signalling molecules, miRNAs, etc. [156,157,158,159]. 

#### 5.1.2. Meiosis

During meiosis, spermatocytes reduce the chromosome copy number from diploid to haploid round spermatids through two consecutive divisions. Interestingly, meiotic cell division occurs in stage XII in mice, but in rats, it is not confined only in a single stage. When B spermatogonia divides by mitosis forming two preleptotene spermatocytes, it represents the meiotic entry, prior to chromosome segregation. They undergo meiotic prophase I, which consists of the reorganization of nuclear architecture and chromosome structure, leading to the formation of bivalent chromosomes through the formation of the synaptonemal complex (SC) or related structures to perform meiotic recombination. Primary spermatocytes go through the preleptotene stage, when DNA replication occurs, and later, undergo the leptotene, zygotene, pachytene, and diplotene stages of prophase I. In mice and humans, the main components of prophase I are conserved, but the organization of axial element (AE) and SC displays sexually different properties, since prophase I of meiosis in males presents pachytene checkpoint and meiotic sex chromosomal inactivation (MSCI). MSCI are accompanied by significant changes in gene expression and modifications to the epigenetic landscape. It occurs when the transcription of most genes on the sex chromosomes is suppressed during meiosis due to the lack of synapsis between the X and Y chromosomes outside the pseudoautosomal regions, localizing the sexual chromosomes in the sex body or XY body. After meiosis, sex chromosome inactivation is predominantly preserved in round spermatids within a heterochromatic structure called post-meiotic sex chromatin (PMSC). However, certain genes bypass this silencing for specific activation. During meiotic prophase, CDKs, cyclins, and non-cyclin CDK activators, are important effectors. That includes, among many others, cyclin A1, Cdk2, and Cdk4. Also, meiosis initiator (MEIOSIN) and STRA8 regulate gene activity by binding to transcription start sites (TSSs) of a wide array of genes involved in meiosis and other spermatogenic processes; many target genes of MEIOSIN and STRA8 are associated with meiotic prophase processes, such as chromosome dynamics and recombination. Then, the first meiotic division produces secondary spermatocytes, which quickly proceed through the second meiotic division, resulting in the formation of haploid round spermatids. Mistakes in these crucial processes can result in aneuploidy and genetic instability, making it critical for them to be regulated by multiple surveillance mechanisms [141,160,161,162,163,164]. 

#### 5.1.3. Testis Cells and Seminiferous Tubule Structure

Within the seminiferous tubule, Sertoli cells, which are polarized epithelial cells, generate the tight BTB, to divide the seminiferous epithelium into basal and luminal compartments [165] (Figure 1). The BTB structure is created by tight junctions, ectoplasmic specializations, desmosomes, and gap junctions that are present between Sertoli cells but not between Sertoli and germ cells or between germ cells in vivo [154]. This barrier is vital, as it protects sperm production from autoimmune reactions [17]. Sertoli cells are the major contributions to the stem cell niche; although, there are also contributing peritubular myoid (contractile cells surrounding seminiferous tubules playing a crucial role in propelling spermatozoa from the seminiferous tubules into the epididymis) and Leydig cells (located in the interstitial space and secreting testosterone in the presence of LH) (Figure 1). Also, testosterone is needed for the maintenance of the BTB, spermatogenesis, and fertility, and it promotes both Sertoli–germ cell junction assembly and disassembly [137,154]. During cellular transformation, germ cells migrate from the basement membrane to the tubular lumen of seminiferous tubules [4]. This transport during the epithelial cycle is possible due to the coordination between the microtubule (MT) and F-actin-based cytoskeletons in Sertoli cells and at the Sertoli–germ cell interphase. Studies in mammalian cells displayed that MTs serve as tracks for carrying several cargos (cellular organelles including proteins, mRNA complexes, endosomes, mitochondria, and cell nuclei), requiring the motor proteins as dyneins in the MT-minus-end-directed transport and others as kinesin, which is responsible for the MT-plus-end-directed transports of cellular organelles. Motor proteins are probably recruited by developing spermatids in order to move up or down the seminiferous epithelium at different stages of the epithelial cycle. Sometimes dyneins and kinesins are attached simultaneously to the same cargo, and there are several protein kinases, phosphatases, and Rab GTPases regulating the activity of motor proteins. In this scenario, the polarized F-actin microfilaments are also involved, which also serve as the tracks to support cellular transport serving as a guide for MT growth [166,167]. 

#### 5.1.4. Spermiogenesis

The spermiogenesis corresponds to the post-meiotic phase or cytodifferentiation, in which haploid round spermatids are going to differentiate into spermatozoa by displaying morphological transformation, including the formation of the flagellum and acrosome (with the contribution of Golgi apparatus), nuclear condensation being histones replaced by protamines and cytoplasmic shedding [141,159,161]. When spermatogenesis is finished, it is followed by spermiation when immotile spermatozoa are released from testis; therefore, they still need to mature throughout the epididymis, followed by capacitation and an acrosome reaction in the female reproductive tract [168]. 

During this late stage, intercellular bridges gain some importance, since round spermatids are haploid, and some individual cells would lack single copy genes, such as those present in sex chromosomes. Therefore, haploid spermatids could behave as functionally diploid by sharing gene products across intercellular bridges. However, the presence of multinucleated spermatids is a hallmark of defective spermiogenesis. Regarding the development of the acrosome, it starts in round spermatids shortly after meiosis, formed by the trafficking of the Golgi-derived vesicles to the nuclear membrane. Then, it gradually extends across the nuclear surface, eventually covering up to half of the anterior portion of the sperm head. For instance, failure in acrosome formation is linked to the known condition, globozoospermia. Moreover, the species-specific sperm head shape is established after the round spermatid nucleus shifts to one side of the cell and gradually transitions from a spherical shape, as nuclear condensation and sperm head shaping begin. Abnormalities in head morphology could be due to problems associated with the formation of the manchette (microtubule-based structure vital for sperm head shaping) or to the chromatin compaction and protamination. Besides, early round spermatids start the assembly to the central microtubule-based component of the flagella, known as the axoneme, which is structured with a central pair of microtubules encircled by nine outer doublet microtubules, forming the characteristic “9 + 2” arrangement. Dynein motors attached to the outer doublets create the forces necessary for antiparallel sliding, which drives the flagellum’s wave-like motion. After the initiation of axoneme formation in early spermiogenesis, secondary structures required for flagella function are assembled during the elongation phase of spermiogenesis. Defects in all these complex processes could lead to issues in flagella assembly and in sperm motility. In the elongation phase of spermiogenesis, the round spermatid nucleus and acrosome becomes polarized to one side of the cell. At this stage, the spermatid connects with a specialized type of adhesion junction known as the ectoplasmic specialization. Defects in the positioning of sperm heads inside the epithelium are probably due to defects in either the ES and/or its ability to be translocated along Sertoli cell microtubules. The last step of spermiogenesis is spermiation, the process in which elongated spermatids undergo final remodelling and are released from the seminiferous epithelium into the lumen of the tubule, preparing for their transport to the epididymis [169,170]. It has been described that sperm had a greater mid-piece volume from polygamous primate species than sperm of a monogamous species, where sperm competition would be higher [171]. 

#### 5.1.5. Mechanisms for Maintaining DNA Integrity in Spermatozoa

To address DNA damage in spermatozoa and preserve genomic integrity, five repair mechanisms have evolved, as follows: nucleotide excision repair, base excision repair, mismatch repair, double-strand break repair, and post-replication repair [172]. The ubiquitin proteasome system is important in every step of spermatogenesis, since deubuquitinating enzymes might be involved in histone removal and protein turnover during meiosis [173]. Several studies on cultured mammalian cells indicated that the ubiquitin–proteasome pathway plays a crucial role in several cellular processes, including DNA repair, protein translocation, circadian rhythm regulation, protein folding, transcription, and apoptosis [172].

#### 5.1.6. Importance of Apoptosis in Spermatogenesis

Apoptosis plays a crucial role in ensuring the proper function and development of male germ cells, from the initial stages of gonadal differentiation in the early embryo to the process of fertilization [174]. That removal of excess germ cells from testicular tissue is crucial in order to control the number of spermatogenic cells that are supported and nourished by the Sertoli cells, since it has been described that up to 75% of germ cells are lost during the development of spermatogonia, ensuring the elimination of those that carry DNA mutations or genes with defects [4]. Mitochondrion is a key intracellular organelle involved in the regulation of apoptosis, which is translationally active during spermatogenesis. This organelle is participating in multiple roles, including ATP production, the generation of reactive oxygen species (ROS), calcium signalling, and apoptosis. Mitochondrial defects are associated with various physiological dysfunctions, including infertility. In the mid-piece of mature mammalian spermatozoa, there are 72 to 80 mitochondria [166]. While mitochondrial proteins play a major role in the activation and termination of intrinsic apoptosis in response to oxidative stress, sperm cell extrinsic apoptosis is mediated by Fas protein receptors [172]. Apoptotic makers possibly related to male infertility would include the activation of phosphatidylserine mark by caspases (early apoptotic marker), DNA fragmentation (late-stage apoptosis), and low motile sperm, which reflects the high presence of apoptosis markers compared to the highly motile sperm cells [172]. 

## 6. New Discoveries and Future Perspectives

### 6.1. Advancements in the Study of Male Hormonal Reproductive Disorders

Fertility issues affect 9% of couples worldwide, and 50% come from the male side, according to the World Health Organization (WHO). Several factors are involved in male infertility, ranging from genetic mutations to lifestyle habits. Impairments at a hormonal level during the cycle of spermatogenesis will result in failure to give rise to suitable spermatozoon. Therefore, fertility in the male side will be compromised. During puberty, adequate spermatogenesis and sperm production are brought about by the role of gonadotropins that lies mainly in generating the cohort of Sertoli, Leydig, and germ cells for adult life. Consequently, if hormone deprivation occurs during this stage of life, scrotal descent and testis development will be impaired; whereas, in adults, only the germ cell composition in somatic cells will be functionally hindered, mainly in Sertoli cells [12].

The most prevalent male genitalia congenital disorder is cryptorchidism. Involved in this disease is INSL3, which participates in the coordination of the descent of the testis during foetal development. It has been reported [175] that knockout male mice for *Insl3*, although viable, show bilateral cryptorchidism caused by abnormalities in the development of the gubernaculum. This structure is a foetal ligament that is attached to the scrotum and caudal epididymis and follows atrophy after birth [176]. This knockout model has been reported to be especially useful for the study of oestrous cycle deregulation in female homozygotes [175]. 

Regarding HPG axis impairments, the role of kisspeptin in the HPG axis has been described, and its potential use as a diagnostic tool and treatment for some reproductive disorders, such as hypogonadism, central precocious puberty (CPP), and female infertility, has been highlighted. Kisspeptin acts as a critical factor regulating GnRH release. In the male side, it is also involved in reproductive behaviour, Leydig cell regulation, spermatogenesis, and sperm function [177]. Rat studies by Ishikawa et al., 2018 [178] and Matsui et al., 2012 [179] have elucidated a great anti-tumour effect in the kisspeptin analogue, TAK-448, which strongly inhibits the HPG axis to a greater extent than GnRH analogues. Therefore, these findings constitute a potential and novel approach in androgen deprivation therapies, indicated for sex hormone-dependent malignancies, such as prostate cancer.

Many challenges are encountered in clinical practice regarding reproductive disorders. However, current knowledge on endocrine dynamics and future advancements will shed light on a better understanding of hormone-dependent reproductive malignancies.

### 6.2. The Testicular Renin–Angiotensin System (RAS)

The latest knowledge about hormone influence in male fertility describes events related with the testicular renin–angiotensin system (RAS). RAS is a crucial intrinsic hormonal pathway within the testes, consisting of key components, such as renin, angiotensinogen, angiotensin-converting enzyme (ACE), and angiotensin receptors. In this localized system, the RAS regulates vital processes, such as steroidogenesis, spermatogenesis, and sperm function, through autocrine and paracrine signalling mechanisms, making it essential to maintain male reproductive health [180]. 

Disruptions in the testicular RAS have been implicated in male infertility, affecting various aspects of sperm metabolism and fertilization. For example, testicular ACE plays a significant role in regulating sperm function by influencing energy production. Consequently, alterations in ACE activity could lead to impaired sperm motility and reduced fertilization potential [181]. Additionally, ACE2, present in the testes, serves a counter-regulatory function by converting angiotensin II into angiotensin (1–7), which may protect sperm from oxidative stress. Dysfunction of this axis could adversely affect sperm quality, further contributing to fertility issues [182]. Moreover, the expression of angiotensin receptors, particularly AT1R, is closely associated with spermatogenesis. The altered expression of these receptors can disrupt normal sperm development, leading to suboptimal sperm function [183].

These findings highlight the critical role of a properly functioning testicular RAS in preserving male fertility. The dysregulation of this system can result in significant reproductive impairments, emphasizing the need for further research to understand its complexities. Ultimately, a deeper understanding of the testicular RAS could aid in the development of targeted therapeutic strategies for addressing male infertility.

### 6.3. Insights in Male Contraception

Male hormonal contraception poses an effective and reversible alternative to female contraception methods, allowing men to share the burden of this issue with their female partners. Therefore, it has potential to positively and significantly have an impact in society.

The exogenous administration of suppressing hormones for both LH and FSH leads to low T levels in the testis, resulting in a significant decrease in sperm count in the ejaculate. Some of these suppressing hormones that can be used are T alone or in combination with progestin or GnRH analogues (Figure 2), and their effect is reversible [184,185]. Although these options are effective and prevent fertility in most men, they are not yet available or approved. These treatments entail some adverse effects regarding sexual drive, acne, and serum cholesterol. Moreover, it is uncertain which side effects would be faced after long-term high doses of T regarding some cancers, such as prostate cancer, and some cardiovascular diseases.

Additionally, the optimization of dosing and delivery are required to overcome limitations and reduce the side effects of these contraceptives. Some of these limitations are the variability in suppression rates regarding different populations, the long duration of the treatment (6–8 weeks due to the 72-day process of spermatogenesis), the risk of pregnancy due to sperm rebound in some men, and a lack of funding from the pharmaceutical industry. To overcome some of these hurdles, novel androgens with dual androgen–progestin action, such as dimethandrolone undecanoate and 11-ß-methyl-19-nortestosterone-17-ß-dodecyl carbonate, have been reported as suitable candidates to be developed as oral pills or long-acting injections. Other promising methods of delivery under research are transdermal gels and implants, providing the user with a more independent and user-friendly alternative [186,187]. 

Aside from male contraception, the mechanism of bypassing the action of GnRH by administering exogenous T, progestins, or GnRH analogues could represent a valuable mechanism to be studied as a therapy for some diseases in which the HPG axis is impaired in males, for example, in congenital hypogonadotrophic hypogonadism (CHH), causing delayed puberty. In addition, several scientific studies have proposed heat stress as a male contraception method [188,189,190] through germ cell apoptosis, inducing reversible oligospermia or azoospermia. Nevertheless, male thermal contraceptive methods (MTC) include the generation of artificial cryptorchidism, heat shock by ultrasound, or local hot water applied to the scrotum. An experiment with voluntary patients of normal semen parameters was performed to analyse the inhibition of spermatogenesis by artificial cryptorchidism, as a contraceptive method in humans, for 12 months [191]. Most of the parameters analysed dropped significantly from the first month, reaching the lowest values from the sixth month on. Ultrasound has been used as a reversible suppressant of spermatogenesis in different species, since it produces a double effect, heat and mechanic, affecting sperm, spermatids, and primary and secondary spermatocytes but Sertoli cells and stem cells, with any kind of genetic damage [192]. 

### 6.4. Advancements in the Gene Regulation of Spermatogenesis

The integration of omics technologies and tools is transforming our understanding of genetic regulation in spermatogenesis. Recent studies have identified previously uncharacterized genes as essential for meiosis, such as *Zcwpw1*, based on high-resolution proteomic dynamics and new machine learning technologies to predict key meiosis proteins [73]. These new predictions require validation, paving the way for further investigations and characterizations of the genes and proteins involved.

Single-cell RNA sequencing (scRNA-seq) has revolutionized transcriptomics by enabling detailed gene expression analysis at the single-cell level. scRNA-seq has emerged as a powerful method to analyse thousands of cells simultaneously and their intercellular relationships, uncovering cellular diversity and intricate molecular mechanisms that were undetectable with the traditional bulk method. By characterizing cellular heterogeneity at the single-cell level, scRNA-seq provides crucial insights into intercellular communication within the testis, helping to map the signalling pathways that regulate spermatogenesis. Moreover, these findings have potential applications in male infertility treatments by improving our understanding of germ cell biology [193,194]. However, one limitation of scRNA-seq is the loss of spatial context, as individual cells are dissociated before sequencing. New spatial transcriptomics approaches are complementing scRNA-seq by adding a spatial context, enabling the study of organized gene expression data within tissue architecture and cellular interactions. Combining scRNA-seq with orthogonal validation methods, such as the co-localization of specific proteins in fixed tissues, enhances the reliability of findings and enables gene expression profiling within its native environment. [195,196]. 

scRNA-seq has been instrumental in dissecting the transcriptional dynamics of spermatogonial differentiation. Key regulatory genes are a main study focus, as this approach can provide deeper insights into the molecular mechanisms behind germline development. Green et al. [193] used scRNA-seq to study transcriptome-wide dynamics during SPG differentiation focusing on selected genes known to be involved in regulation, such as *Sohlhs*, *Dmrts*, *Kit*, and *Stra8*. The already described marker genes can be targeted to understand their expression. Additionally, new spatial transcriptomics approaches are complementing scRNA-seq by adding a spatial context, enabling the study of organized gene expression data within tissue architecture and cellular interactions [196]. Through scRNA spatial transcriptomics, marker genes can also be targeted in order to find spatial expression patterns. *Piwil4* (piwi-like RNA-mediated gene silencing 4) and *Etv5* have been found to be expressed in different human spermatogonial developing states [39]. Moreover, a recent scRNA-seq study on SCARKO mice, which lack AR in Sertoli cells, highlights AR’s crucial signalling role in spermatogenesis. AR deficiency in Sertoli cells leads to a disruption in the regulation of spermatogenesis-related genes, triggering apoptosis and impairing germ cell development, as it is essential for maintaining Sertoli cell functions, proper gene expression profiles, and the correct progression of spermatogenesis [197]. 

Beyond marker genes, scRNA-seq has also enabled the identification of distinct spermatogonial states and regulatory networks that regulate their transitions. This way, scRNA-seq has been used to characterize SSCs states during spermatogenesis in adult males. It allowed the identification of five distinct SSC states, distinct marker genes, and transcription factor expression in each state, revealing their molecular regulation during self-renewal and differentiation [39,195,198,199]. Multiple SSC states mean that spermatogonial differentiation is a gradual, dynamic process, which is crucial for understanding the regulation of stem cell proliferation and differentiation [200]. Further studies using scRNA-seq will be essential to deepen our understanding of these processes and uncover the molecular mechanisms underlying SSC behaviours in various contexts. Furthermore, integrating additional techniques with scRNA-seq can facilitate a more comprehensive analysis of the epigenome and help identify potential transcriptional regulators. For example, in H.Q. Wang et al. [201], authors describe a transcriptional reprogramming process during spermatogenesis, between the zygotene and pachytene stages, in both humans and mice. The scATAC-seq method enables the detection of changes in chromatin accessibility at a single-cell level, providing insights into which genome regions are being actively regulated during the zygotene–pachytene transcriptional alteration process (ZPT). They identified 282 transcriptional regulators with changes in motif accessibility during ZPT. For previously reported TRs, such as CREM, this suggests potential involvement in regulating the transition.

As multi-omics knowledge continues to increase, supported by techniques such as scRNA-seq, newly identified gene candidates could be used as biomarkers for infertility diagnostics. *Rbm46* is an example, as it has been recently found to be a key element in spermatogenesis regulation in order to properly exit mitosis during meiosis [202]. Its downregulation in azoospermic patients suggests that *Rbm46* could be a key factor in male infertility. Studies in zebrafish models have also revealed that depletion of *RBM46* disrupts essential genes, such as *Dazl*, *Nanos3*, and *Sycp3*, further confirming its importance in germ cell development and meiosis [202]. These findings make Rbm46 a promising candidate for future research into male infertility, opening the door for investigations in other animal models, like mice, ultimately paving the way for clinical applications in humans. miRNA, lncRNA, and circRNAs are also novel gene candidates; their expression profiles in testicular tissues have been increasingly studied and found to be differentially expressed in OA and NOA patients [60,66,68]. This significant differential expression in patients with NOA make them another potentially non-invasive therapeutic, molecular biomarker and drug target. Using the advancements of gene editing technologies, such as tools like CRISPR/Cas9, has enabled the manipulation of specific genes to investigate their function, paving the way for gene therapies to treat male infertility. The broad knockout approach has evolved into more targeted, specific knockouts, enabling studies to gain a more detailed and comprehensive understanding of spermatogenesis [203]. That is why clinically, non-coding RNAs are emerging as promising biomarkers for diagnosing reproductive disorders and as targets for innovative treatments [60,204]. Dysfunctions in spermatogenesis, which are responsible for male infertility, often have a genetic basis, making CRISPR/Cas9 gene editing a promising therapeutic approach. Previously mentioned miR-202, a target microRNA for male infertility because of its crucial role in SSC regulation, was deleted using CRISPR/Cas9 for additional functional analysis [205]. Gene therapy and new technologies can be used to treat male infertility caused by mutations in key genes for sperm development or to correct epigenetic imbalances. As an example, male infertility has also been restored in mice using CRISPR/Cas9, targeting mutated marker genes that lead to impaired spermatogenesis and, consequently, infertility, such as *Kit* [205] and *Tex11* [206]. 

As seen in Liang et al. [69] a deeper understanding of the molecular pathways regulating spermatogenesis can lead to significant advancements in potential therapeutic approaches for male infertility. A promising avenue for future research is the clinical application of identified spermatogenesis regulators. The direct involvement of *Dazl* [41], *Plzf* [32], and miR-202 [207] in infertility and spermatogenesis dysfunctions has been well-documented in the literature, yet limited research in infertile patients has been conducted due to ethical concerns, technical challenges, and the complexity of accurate spermatogenesis-related disorder models in clinical settings. However, advances in gene editing technologies, such as CRISPR/Cas9, combined with in vitro models and personalized medicine, offer promising avenues to overcome these challenges and enable the further exploration of spermatogenesis in human patients. This could provide insights into new therapeutic strategies, where, in the future, the development of RNA-based therapies and the combination of genetic and epigenetic approaches have the potential to redefine male infertility treatment, allowing for more targeted and personalized interventions.

### 6.5. Advancements in In Vitro Spermatogenesis

Testis is a complex organ, where different cells interact to achieve healthy sperm production. Several factors can compromise this process, leading to fertility failure. Efforts to study it in vitro have been ongoing for decades, driven by the need to better understand male infertility, reproductive toxicology, and potential regenerative medicine applications.

Early studies used 2D monolayer cultures to isolate testicular cells, such as spermatogonia, Sertoli cells, and Leydig cells. While these approaches provided valuable insights into individual cell behaviours, they lacked the microenvironment and spatial organization necessary for full spermatogenesis. The main limitations of this culture approach hinder its ability to accurately mimic the seminiferous tubule structure and the inability to maintain germ cell differentiation beyond a certain stage [208]. Alternatively, the in vitro culture of testicular tissue slices or fragments has been a significant advancement. These explants retain the architecture of the seminiferous tubules, enabling better cellular interactions. A 2011 study by Sato et al. demonstrated complete in vitro spermatogenesis in neonatal mouse testicular tissue cultured with optimized conditions, producing functional sperm capable of fertilization [209]. The limitation of this technique is the limited long-term viability of tissue cultures and the difficulty in translating protocols to adult human testicular tissue. At this point, organoids offer a promising solution by providing an in vitro model that mimics in vivo processes [210]. Testicular organoids are 3D cell culture systems that self-assemble from testicular cells, creating a microenvironment more representative of in vivo conditions. They can mimic the seminiferous tubule structure and facilitate cellular interactions and signalling pathways critical for germ cell development. With this technique, partial spermatogenesis in mouse organoids has been achieved, supporting further germ cell differentiation by the introduction of growth factors, like FSH, testosterone, and retinoic acid. The main challenge is, on one hand, the difficulty in achieving complete meiosis and the production of fully mature sperm and, on the other hand, the variability in organoid formation and functionality. Organoid techniques rely on the knowledge of induced pluripotent stem cells (iPSCs) and embryonic stem cells (ESCs) that have been differentiated into primordial germ cell-like cells (PGCLCs) and even spermatogonial-like cells. In 2016, a study by Q. Zhou et al. [211] achieved complete in vitro spermatogenesis from PGCLCs derived from mouse ESCs. Moreover, microfluidic platforms are a new technique that recreate dynamic environments with precise control over nutrient flow, temperature, and hormone delivery. This allows us to improve germ cell survival and differentiation and model testicular toxicology with high reproducibility.

The combination of these techniques amplifies the options to design a successful protocol to study this complex process.

So, research in this area is trying to develop robust and reproducible protocols for large-scale in vitro spermatogenesis, achieving functional sperm production from germ cells as the ultimate goal. Combining in vitro testicular systems with ovarian organoids or other reproductive models will allow for an advancement in the knowledge of broader aspects of fertility. In addition, it could be useful for clinical applications, such as the generation of sperm for individuals with non-obstructive azoospermia or for preserving the genetic material of endangered species.

### 6.6. Environmental Threats for Male Fertility

Between 1970 and 2020, fertility rates declined across all countries globally. Although the world’s population continues to grow, the rate of this growth is decreasing. However, this situation seems to be different depending on each country, but still, across men from all continents, the average sperm concentration dropped by 51.6% between 1973 and 2018, with around 7% of all men affected by male infertility all over the world. The widespread presence of environmental stressors has become a significant contributor to the increasing incidence of male infertility and a reduction in sperm quality worldwide [212,213]. 

The harmful effect of environmental factors on male fertility (Figure 6) could be explained by the effect in sperm of ROS, the impact of epigenetics, and genetic susceptibility. One of these factors is xenobiotics, such as dioxins and furans, from the transport industry, among others; bisphenol A (BPA), present in manufactured plastics, cosmetics, and as an antioxidant in the food industry; phthalates, to give flexibility to plastic; micropollutants and nanoparticles from nanotechnologies (such as silver, copper oxide, zinc oxide, cobalt, titanium oxide, nickel oxide) that ends up in the aquatic environment, including pharmaceuticals and sex hormones; alcohol; and pesticides. Interestingly, xenobiotics behave as endocrine disruptors, interfering in vertebrate species with both genomic and non-genomic pathways (Gallo et al., 2020). Other factors are air pollution (sulphur oxides, nitrogen oxides, carbon oxides, dust, soot, and ashes), heavy metals (such as lead, cadmium, and mercury), ionizing radiation, or occupational dust [212,213,214]. Lifestyle factors (such as smoking, alcohol consumption, and BMI), health conditions (such as varicocele, impaired glucose tolerance, infections, and some malignancies), and advanced paternal age have shown an effect in sperm DNA fragmentation [215]. However, the quality of semen could be improved by following a healthy diet rich in plant-based foods, such as Mediterranean or vegetarian diets or a dietary approach to stop hypertension (DASH) [216,217]; although, more research in the field is needed.

Several studies pointed out that the prolonged use of and exposure to mobile phones [218] and Wi-Fi [219] and, therefore, to RF-EMR (radio frequency electromagnetic radiation), which is a non-ionizing radiation, could have adverse effects on human health, including spermatogenesis [218]. The spermatogenesis issues could be due to thermal stress increasing the testicular temperature and due to an oxidative stress increasing ROS and, therefore, DNA damage. Both of them would lead to decreased sperm counts and maturation, a reduced epididymal weight, a reduction in motility, and alterations in morphology and cell viability [220]. 

Furthermore, reproduction is also affected by the two direct consequences of climate change due to the rise in greenhouse gas emissions—global warming (increase in global temperature) and ocean acidification [208]. Temperature increases exert a negative effect in sperm fitness, as has been previously described in this work; ocean acidification also shows a negative impact on the reproduction of the majority of marine species, since they are external fertilisers and release sperm and eggs into the water [221]. 

Therefore, the exposure to adverse environmental factors can lead to reduced semen quality in terms of decreased sperm concentration, impaired motility, viability, and normal morphology, as well as increased sperm DNA fragmentation, mitochondrial dysfunction, and membrane integrity, all contributing to male infertility [212]. 

### 6.7. The Gut Microbiota–Testis Axis

Recent research has highlighted the critical role of the gut microbiota in various processes occurring within the testes, giving rise to the concept of the gut microbiota–testis axis. This axis refers to the bidirectional communication between the gut microbiota and the testes, which significantly influences male reproductive health through a range of biological mechanisms. The gut microbiota, composed of trillions of microorganisms, plays an essential role in metabolism, immune regulation, and endocrine signalling. These microbes produce key metabolites, such as short-chain fatty acids (SCFAs), polyunsaturated fatty acids (PUFAs), and bile acids, which enter the bloodstream and reach the testes, thereby modulating testicular function. This interaction involves complex signalling pathways that regulate steroidogenesis, spermatogenesis, and overall testicular health [222]. Consequently, disruptions in the gut microbiota, a condition known as dysbiosis, can adversely affect fertility.

Dysbiosis has been linked to increased intestinal permeability, allowing lipopolysaccharides (LPS) to enter the bloodstream and trigger immune responses that lead to systemic inflammation. This inflammation can negatively affect sperm production and quality, contributing to male infertility [223]. Additionally, certain bacterial genera, such as Lactobacillus, Bifidobacterium, and Enterococcus, have been shown to promote sperm quality, likely due to their ability to modulate inflammatory responses and produce bioactive metabolites. In contrast, mucolytic bacteria, like Bacteroides caccae and Akkermansia muciniphila, can degrade the intestinal mucus barrier, increasing LPS translocation and causing further inflammation, which impairs spermatogenesis and disrupts testicular health [224].

Recent studies have demonstrated that interventions aimed at restoring gut microbiota balance, such as probiotics, prebiotics, and faecal microbiota transplantation (FMT), can improve both systemic and testicular health, thereby enhancing spermatogenesis in animal models [225]. For example, supplementation with alginate oligosaccharides (AOS) has been shown to modulate the gut microbiota, improving semen quality and the testicular environment, which ultimately enhances fertility in male subjects with obesity-induced infertility [223]. 

These findings underscore the importance of maintaining a balanced gut microbiota for optimal male fertility and suggest that interventions targeting the gut microbiota, such as probiotic supplementation, prebiotics, synbiotics, and FMT, could serve as promising therapeutic strategies to restore the gut–testis axis and improve reproductive health [226]. 

## 7. Conclusions

The complex process of spermatogenesis depends on the proper functioning of multiple events, such as hormonal influence, adequate genetic and epigenetic regulation, and the correct balance between cellular proliferation and differentiation. All of these processes must occur under the appropriate temperature and environmental conditions. Lifestyle, climatic change, and environmental contamination are threats to this delicate process. Recent research on hormone influence allows us to understand the germ cell niche, and the delicate balance of the whole process. Cell-to-cell interaction and the internal cycles of spermatogenesis are orchestrated by multiple hormones assuring the success of the system. Although the main spermatogenesis genes are described, new discoveries in the role of miRNA and epigenetic events complete the knowledge about spermatogenesis regulation, elucidating their role in cell proliferation and differentiation events at the molecular level. Although spermatogenesis has been studied for a long time, new influences and interactions, such as the gut–microbiota testis axis and the testicular renin–angiotensin system, show us that its complexity still requires more research to fully understand and control germ cell development. Due to advancement in cell culture techniques, in vitro spermatogenesis appears to be a promising tool for investigating and understanding this complex process. To achieve this, it is necessary to create a suitable microenvironment that mimics the in vivo conditions of the testis and supports the survival and development of all the cell types involved in spermatogenesis, ultimately achieving complete and functional spermatogenesis. A better understanding of how to maintain somatic and germinal cells in vitro is required [208,227], and new in vitro models, such as organoids, could play a key role in this aim [210,228,229,230]. This strategy offers the potential to simplify cell-to-cell interactions, clarifying the role of hormonal and genetic influence on the process and providing a suitable model for studying the impact of temperature, environmental change, and toxic substances in the spermatogenesis process [231,232,233,234,235,236]. This strategy not only contributes to basic research but also has potential clinical applications that include infertility treatment, fertility preservation, conservation biology, genetic disorder prevention, and research into human germ cell development, amongst others [232].

## Figures and Tables

**Figure 1 biomolecules-15-00500-f001:**
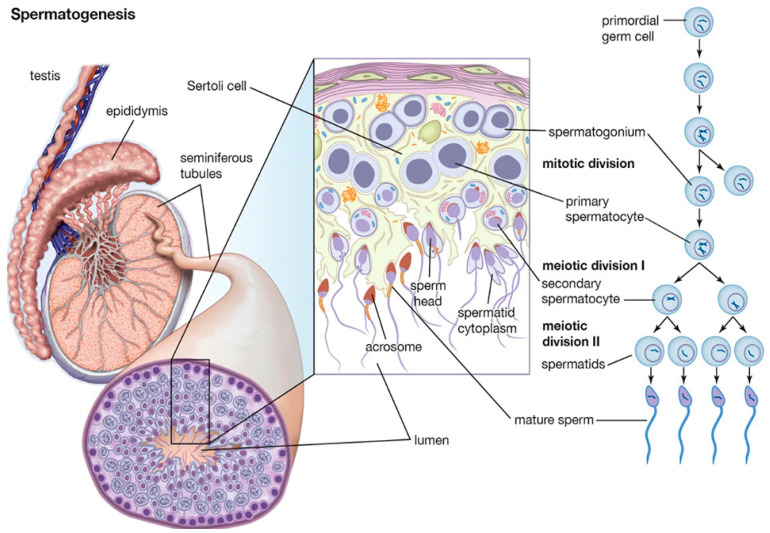
Reproductive male track, showing epididymis and testis structure. Testis are constituted by seminiferous tubules were somatic and germ cells interact to achieve, firstly, mitosis and meiosis divisions and, secondly, the differentiation of germinal cells, through the spermatogenesis process. Scheme courtesy of the Editors of Encyclopaedia Britannica, 2024 [5]; used with permission.

**Figure 2 biomolecules-15-00500-f002:**
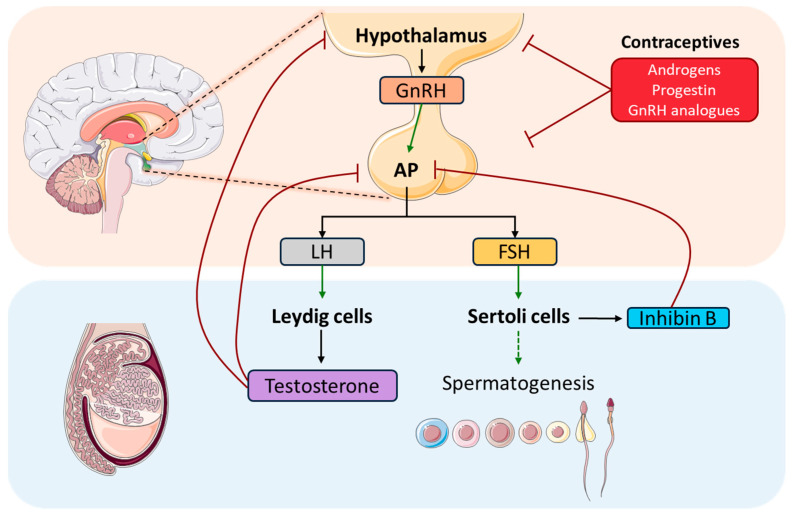
A schematic representation of the endocrine mechanism regulating the male reproductive system, including the inhibitory action of exogenous contraceptives. Gonadotropin releasing hormone (GnRH); Anterior pituitary (AP); Luteinizing hormone (LH); Follicle-stimulating hormone (FSH). Pictures of this scheme are obtained from Servier Medical Art [9].

**Figure 3 biomolecules-15-00500-f003:**
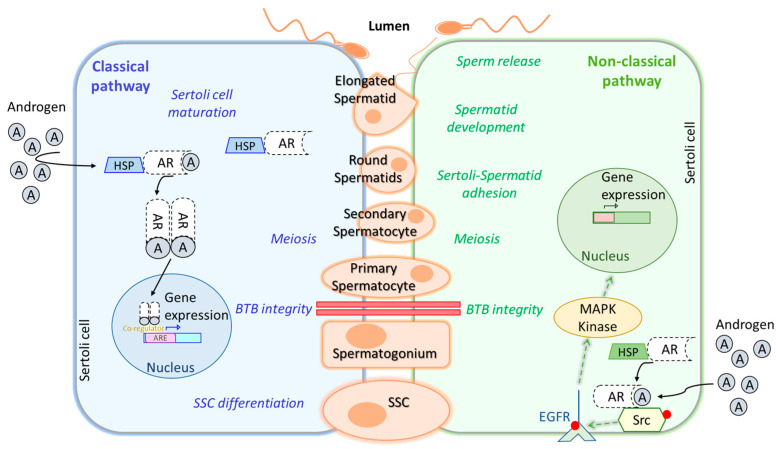
Representation of the classical and non-classical pathways in Sertoli cells. Two contiguous Sertoli cells are supporting Spermatogonial Stem Cells (SSC), spermatogonium, primary spermatocytes, secondary spermatocyte, spermatids (both round and elongated), and spermatozoa, providing the necessary nutrients and an environment for proper spermatogenesis. In the classical pathway, heat shock proteins (HSP) bind to the cytoplasmic androgen receptor (AR), which recognizes and complexes with androgen. Once HSP detaches, AR dimerizes and translocates into the nucleus and binds to the genomic androgen response elements (AREs) of the target genes. This regulates its transcription in order to ensure SSC differentiation, blood–testis–barrier (BTB) integrity, meiotic progression, and Sertoli cell maturation. The non-classical pathway begins with androgen binding to membrane-associated AR, which interacts with Src, leading to the phosphorylation of epidermal growth factor receptor (EGFR). This activation trigger MAPK cascades, inducing transcriptions of genes that regulate integrity of BTB, meiosis, adhesion of Sertoli and spermatids, spermatid development, and sperm release. A Sertoli cell can exhibit both pathways simultaneously.

**Figure 4 biomolecules-15-00500-f004:**
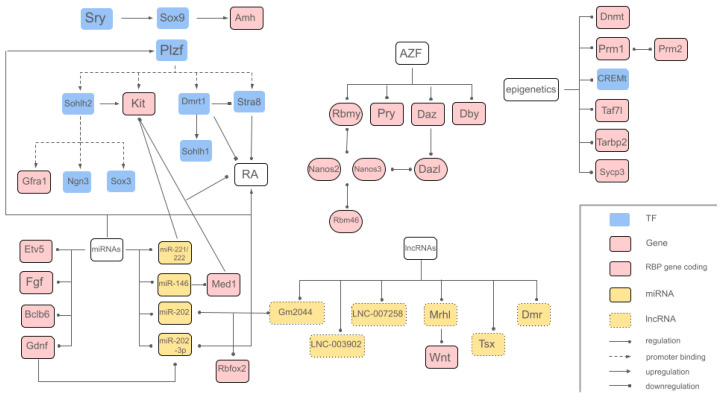
Schematic representation of key genes, transcriptional and post-transcriptional factors, and epigenetic regulators involved in spermatogenesis, along with their interactions in spermatogenesis regulation.

**Figure 5 biomolecules-15-00500-f005:**
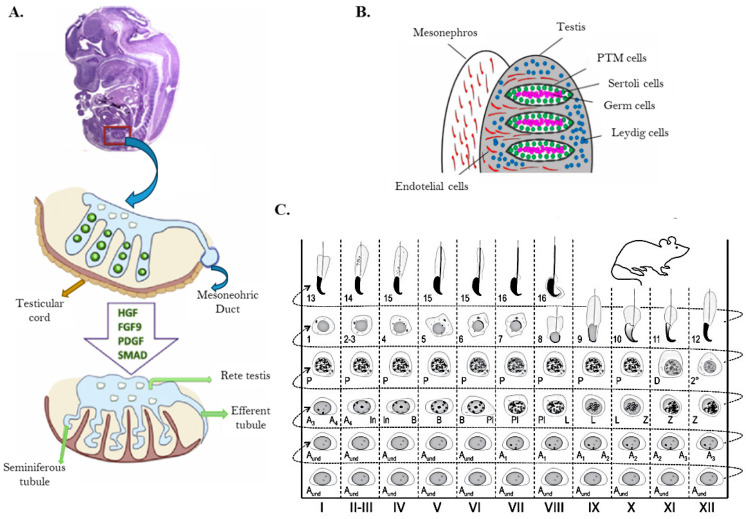
(**A**) Mouse testis development. (**B**) Cell lineages of developing testis: peritubular myoid cells (PTM), Sertoli cells, germ cells, and Leydig cells. (**C**) Seminiferous epithelial cycle map of mouse spermatogenesis: The columns show the different stages of the seminiferous epithelial cycle (marked with Roman numerals I–XII). The progress of germ cell differentiation is represented from (**left**) to (**right**) and from (**bottom**) to (**top**). A_und_, undifferentiated spermatogonia; A1–A4, type A1–A4 spermatogonia; In, intermediate spermatogonia; B, type B spermatogonia; Pl, preleptotene spermatocytes; L, leptotene spermatocytes; Z, zygotene spermatocytes; P, pachytene spermatocytes; D, diplotene spermatocytes; and 2°, secondary spermatocytes plus meiotic divisions. Arabic numerals 1–16 refer to steps of post-meiotic spermatid maturation (spermiogenesis). Reproduced with permission from (**A**) [138]; (**B**) [139] and (**C**) [140].

**Figure 6 biomolecules-15-00500-f006:**
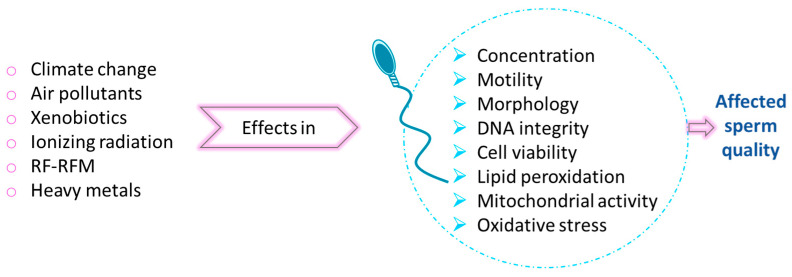
Effects of environmental stressors on sperm parameters and, consequently, on sperm quality and subsequent fertilization capability.

**Table 1 biomolecules-15-00500-t001:** Principal genes, transcription, and post-transcription factors and epigenetic regulators in spermatogenesis.

Type	Elements	Description
**Genes and transcription factors**	*Sry*	Master regulator initiating testicular differentiation; activates *SOX9*
	*Sox9*	Promotes seminiferous tubule formation and maintenance
	*Plzf*	Critical for spermatogonial stem cell maintenance; binds and represses differentiation-related genes
	*Sohlh1*, *Sohlh2*	Spermatogonia regulation by suppresing or inducing related genes
	*Gfra1*, *Ngn3*, *and Sox3*	Directly involved in spermatogenesis regulation
	*Kit*	Regulates spermatogonial differentiation
	*Stra8*	Retinoid acid target; regulates spermatogonial differentiation
	*Dmrt1*	Differentiating marker; prevents meiosis and promotes spermatogonia development
	*Rbmy1*, *Pry*, *Daz*, *Dby*	Directly involved in essential spermatogenesis processes
	*Dazl*	Promotes germ cell development, determination, and meiotic progression
	*Tbx3*, *Utf1*	Regulate spermatogonial stem cell differentiation
	CREMt	Controls post-meiotic germ cell differentiation
**Post-Transcriptional Regulation**	Genes	
	*Nanos2*, *Nanos3*	Encode RNA-binding protein (RBP) to regulate primordial germ cells and germ cell development via mRNA downregulation
	*Tial1*, *Dnd1*	Encode RBP to regulate primordial germ cells
	*Rbm46*	Encode RBP to regulate mRNA translation and stability; essential for meiosis and spermatogenesis
	*Gdnf*, *Fgf*, *Bclb6*, *Etv5*	Promote SSC self-renewal and maintain the niche
	miRNA	
	miR-146	Inhibits RA-mediated differentiation
	miR-221/222	Maintain undifferentiated state
	miR-202-3p	Maintain undifferentiated state
	miR-202	Related to lncRNA Gm2044 and spermatogonial arrest; regulates *Rbfox2*
	lncRNA	
	*Mrhl*	Regulates Wnt and spermatogonia development
	*Tsx*	Regulates pachytene spermatocyte development
	*Dmr*	Negative regulator of *Dmrt1* in male development
	*Rbfox2*	Regulates cell proliferation and differentiation in spermatogenesis
	LNC_003902	Regulation of key genes essential for spermatogenesis and testis development
	LNC_007258	Regulation of key genes essential for spermatogenesis and testis development
	lncRNA Gm2044	Related to miR-202 and spermatogonial arrest; regulates *Rbfox2*
	PiRNAs	Silence transposable elements (TEs) during meiosis
	circRNAs	MiRNA sponges; inhibit miRNA functions and regulate gene expression network
**Epigenetic Regulation**	DNA methylation	
	N6-methyladenosine	Plays a crucial role in gametogenesis
	*Dnmt1*	Essential for imprinting and germline-specific methylation
	*Dnmt3a*, *Dnmt3l*	Novo methylation during prenatal development
	Specialized chromatin structures	
	Protamines	
	*Prm1/Prm2*	Required for meiosis and spermiogenesis to ensure proper DNA condensation and stability
	*Taf7l*, *Tarbp2*	Protamine regulators
	*Sycp3*	Meiotic chromosome compaction

**Table 2 biomolecules-15-00500-t002:** Spermatogenesis length in several species.

Group	Specie	Spermatogenesis Length
**Mammals**	Human	74 days [30]
	Rat	52 days [143]
	Mouse	~35 days [144]
	Sirian hamster	~35 days [145]
	Dog	~61 days [146]
	Bovine cattle	61 days [147]
	Horse	~55–57 days [148]
	Pig	40 days [149]
**Birds**	Chicken	~14 days [150]
**Reptiles**	Ex. Scorpion	53 days [151]
**Fish**	Guppy	36 days [152]
	Trout (seasonal)	Weeks to months [153]

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
