# Peer review of "Mechanisms of Hormonal, Genetic, and Temperature Regulation of Germ Cell Proliferation, Differentiation, and Death During Spermatogenesis"

_biomolecules, 2025, doi:10.3390/biom15040500_

Round 1

Reviewer 1 Report (New Reviewer)

Comments and Suggestions for Authors

This paper summarizes the mechanisms of hormonal regulation, gene regulation and temperature regulation during spermatogenesis, and how these mechanisms affect the proliferation, differentiation and apoptosis of germ cells. However, in order to improve the scientific rigor and readability of the manuscript, several aspects need clarification and improvement.

1.​Scope of research and elaboration of details:This review covers too much content, resulting in the neglect of many details. The author should consider limiting the scope of the review to "normal spermatogenesis" or a certain specific pathological type, such as oligospermia, teratospermia or azoospermia. Moreover, this review is mostly descriptive, merely summarizing the results without sufficient discussion. These discussions should include, but are not limited to: details of the included studies (sample conditions, experimental design, etc.) and their deficiencies, and the interpretation of negative or controversial results. These are two issues that urgently need to be addressed in this review.

2. ​Update of the hormonal regulation section:In the hormonal regulation section, the content mainly summarizes the previous consensus content and fails to include the latest progress, such as the testicular local renin - angiotensin system and the gut microbiota - testis axis.

3. ​Consideration of single - cell sequencing technology:Single - cell sequencing technology has been widely applied in the field of reproduction, but this paper, especially in the genetic regulation part, does not fully consider this technology. It is recommended to add relevant discussions.

Comments on the Quality of English Language

OK

Author Response

Reviewer 2 Report (New Reviewer)

Comments and Suggestions for Authors

In this review, the authors provide an interesting and up-to-date perspective on the critical role of hormonal, genetic and temperature regulation during spermatogenesis. The work is organized in a very coherent and comprehensive form and is very well written. The authors present/discuss in minute detail the hormonal regulation in Spermatogenesis in chapter 2, providing a very inclusive view and also an excellent art work (Figs 1-3) focusing the topic. The same is true for all other chapters of the manuscript. All issues are meticulously aligned and provide a clear picture of the current knowledge of all subjects covered. The chapter on "New discoveries and future perspectives" is an excelent image of the actual knowledge in the matters and of the possibilities it opens in many fields related to reproduction.

Some comments related to some minor typos I found while reading the manuscript. Most are minute details that did not constitute any difficulty in understanding the content of the text.

Minor comments:

  1. Page 3 line 98: “Also is included the inhibitory action from exogenous contraceptives over it.” Consider revising the sentence.
  2. Page 3 line 107: “T”; the first time in the text that Testosterone is mentioned should be provided the full word and not just the abbreviature.
  3. Page 4 lines 117-118; “have suggested”; the way the sentence is written almost refers das the Knockout animal models are the ones suggesting! Should refer that Results from studies involving those animals have suggested, for example!
  4. Page 4 line 122: “counteractors factors”; sounds strange. Should it be instead counteracting factors?
  5. Page 5 line 173: “such as, P450”; no need for the comma.
  6. Page 5 line 228: “It acts limiting RA response”; should it include by between acts and limiting?
  7. Page 8 lines 346-347: “plays a crucial role in gametogenesis across multiple.” Something is missing at the end of the sentence. Multiple what?
  8. Page 8 line 357: “de novo”, should it be italic?
  9. Page 9 line 370: “as it is been previously”; use has instead of is!
  10. Page 12 line 391; the authors refer as title for chapter/section 4 of the manuscript “Temperature regulation”; I am of the opinion that Regulation by temperature would better define what is being presented/discussed.
  11. Page 13 line 427: “leads in a reduction”; change in by to!
  12. Pager 14 lines 485-486: “affects both animals and human infertility”; since affects could either mean improve or deteriorate why mention human infertility and not just human fertility?
  13. Page 14 line 488: “with an increased mortality rates”; either remove an and keep rates or maintain an and change to rate.
  14. Page 14 line 501: “or a general from mammals”; why the need for a? Or should it be instead: a general mechanism from humans?
  15. Page 19 line 665: “And defects in positioning”; why not starting just with the word Defects? There is no need to initiate the sentence with And.
  • Page 19 line 668: “process, in”; no need for comma!
  • Page 22 line 853: “ineffective” should be change to inefficacy or inefficiency.
  • Page 23 line 869: “The main problem that presents”; include is between that and presents?
  • Page 23 line 905: “pesticides…”; the authors provide so man examples that there is no need to use …;

  16. Page 23 line 912: “age, shown”, insert have before shown.

Round 2

Reviewer 1 Report (New Reviewer)

Comments and Suggestions for Authors

In future manuscript revisions, it is recommended to highlight or mark the revised content in red, instead of using the track changes format, as the latter makes the text more difficult to read. For instance, in the revised version, a large number of changes are related to the citation of references.

This manuscript is a resubmission of an earlier submission. The following is a list of the peer review reports and author responses from that submission.

Round 1

Reviewer 1 Report

Comments and Suggestions for Authors

The paper contains comprehensive data on the course and regulation of the spermatogenesis process, mainly in mammals with reference to selected other animal groups. The paper is written in a very interesting, understandable way, and contains a lot of important data based on a very rich (over 140 items) literature. I read it with interest and I am convinced that it will be an article that many readers will be eager to read.

I have no substantive comments; in my opinion, the paper is suitable for printing in its current form.

I have only a few minor comments, mainly editorial:

1. In my opinion, the abstract is too long, discouraging to read it.

2. Figures 2 and 3 are not cited in the text.

3. In Figure 2, the abbreviation PTM cells and the abbreviations and symbols in point C should be explained.

4. Line 360: should be “species” and not “specie”; the same in table 2

5. I suggest using the more commonly used form “gonadotropin” instead of “gonadotrophin” throughout the text.

Author Response

  1. In my opinion, the abstract is too long, discouraging to read it.

Agreeing with reviewer, abstract has been reduced. Lines from 10 to 31.

  1. Figures 2 and 3 are not cited in the text.

We have cited the figures in the text. Line    (Fig.2) and line 135 (fig. 3).

  1. In Figure 2, the abbreviation PTM cells and the abbreviations and symbols in point C should be explained.

Figure explanation has been improve to offer better understanding of the content. It has been included abbreviations and symbols in point Fig 5C (Figure number has been changed because of the incorporation of the new figures). Lines 508-518.

  1. Line 360: should be “species” and not “specie”; the same in table 2

It has been corrected. Line 348 and line 530 (Table 2).

  1. I suggest using the more commonly used form “gonadotropin” instead of “gonadotrophin” throughout the text.

It has been changed, and used the common form “gonadotropin” instead of “gonadotrophin” Lines 81-89, 190, 713.

Reviewer 2 Report

Comments and Suggestions for Authors

This article elaborates on the complex and highly regulated process of spermatogenesis, including germ cell proliferation, differentiation, and apoptosis, and how these processes are regulated by hormonal, genetic, and environmental factors, especially temperature. This review highlights the key role of hormonal regulation in spermatogenesis and explores the impact of epigenetic modifications, such as DNA methylation, histone modifications, and chromatin remodeling, on germ cell proliferation and differentiation. In addition, the importance of temperature regulation in mammalian spermatogenesis and the potential impact of heat stress on sperm quality and male fertility are discussed, providing a new perspective on the etiology of male fertility decline and male infertility. However, there are several aspects that require further refinement or clarification by the authors before submitting the manuscript for publication.

1. The article attempts to summarize all events in spermatogenesis but fails to delve deeply into any one event or process, with most of the content being a summary of known and outdated knowledge. Regarding a particular process, much of the content is incomplete, for example, why only 8 genes and transcription factors are listed in table 1, which is clearly incomplete; thus, it needs to be explained what makes these 8 factors special? The illustrations also fail to incorporate the latest developments in the field. In addition, the content about contraception and in vitro spermatogenesis is even more confusing, leaving readers unsure of the focuses of this article.

2. The section of "Genetic regulation in spermatogenesis" involves the regulation of many genes, transcription factors and non-coding RNA on spermatogenesis. Biorender, Photoshop and other mapping software can be used to make regulatory maps with latest findings.

3. To further investigate the molecular mechanisms of how specific hormones regulate germ cell proliferation and differentiation, as well as the signaling pathways and downstream target genes of these hormones, in order to reveal the precise mechanism of action of these hormones in spermatogenesis.

4. In order to demonstrate the importance of environmental changes in spermatogenesis, it is recommended to increase the influence of environmental factors other than temperature (e.g., radiation, chemicals, etc.) on spermatogenesis.

5. Table1 and 2 should use the three-line table.

6. Although various regulatory factors are mentioned, the elaboration on some key mechanisms is not in-depth enough. For example, in the hormone regulation part, only a general description is given on the molecular signaling pathways by which FSH and LH precisely regulate germ cell development, without in-depth exploration of its complex network regulatory mechanism and the details of upstream and downstream signal transduction. In fact, the hormones that influence spermatogenesis encompass a spectrum beyond FSH, LH, and Testosterone, yet, the manuscript fails to elucidate the roles of other hormones that contribute to this process.

7. The article provides a detailed account of the repercussions and hazards of heat stress, which, given the escalating concern over its impact on male fertility, necessitates the provision of examples illustrating the specific scenarios that may precipitate such thermal stress.

8. The prospective section alludes to applications in contraception; could there be a more profound investigation into how the pivotal molecular mechanisms within spermatogenesis might inform therapeutic strategies for male infertility and facilitate the identification of novel therapeutic targets?

9. The review is mainly based on the summary and collation of existing literature and lacks in-depth analysis and integration of original experimental data. Thus, it is difficult to deeply reveal the internal connections and dynamic changes of regulatory mechanisms at the data level.

10. The future outlook section proposes multiple research directions. However, for how to achieve these goals, such as which genes (examples) should be specifically targeted and what strategies should be adopted in gene therapy, it does not provide a clear roadmap and actionable suggestions, reducing its guiding value for subsequent research.

Reviewer 3 Report

Comments and Suggestions for Authors

The authors propose the review in relation to the various regulatory mechanisms underlying the kinetics and development of spermatogenesis in mammals, including humans, that are known to date.

My decision, with respect to this article is that it is not acceptable for publication in your journal. Below I indicate some of the elements that in my opinion are sufficient for the review to be either completely modified or given another focus or perhaps placed in another journal:

1.- The review is not innovative, since many reviews are known today in relation to the different factors that regulate the progression and establishment of spermatogenesis in mammals.

2.- The hormonal mechanisms are very poorly developed, and what is described in the article is very elementary.

3.- I consider that if the title emphasizes those hormonal mechanisms that are regulating spermatogenesis, both the text and the figures that were incorporated are too brief in their development and explanation. Except for Table 1, which is not even indicated in the text, all the figures included are very deficient in their construction and layout. The truth is that images of specific histological sections of cross sections of a non-primate mammal testicle should have been included, as well as some human biopsy material, to describe and demonstrate the main elements that constitute the histological organization of this organ, especially at the level of the seminiferous tubule and the interstitial tissue. This would greatly help to understand the context in which the authors describe the different regulatory factors and mechanisms that are involved in this very finely controlled process.

4.- I get the impression that the review was not thoroughly and rigorously reviewed by the authors, as there are many writing and editing errors and some concepts that were not adequately described, for example, on line 107, it is mentioned that Leydig cells are included in the seminiferous tubule, which I consider to be a conceptual error that cannot go unnoticed. Also on line 323, a paragraph is written, which I think has little to do with what was being discussed before. What is described is valid, but in my opinion a better link was not established between what was discussed before and this paragraph.

5.- On the other hand, in relation to testosterone, very little is discussed about the true role that it has in maintaining spermatogenesis and practically very little or nothing is discussed about the role that the androgen receptor has in the mechanisms that regulate the maturation and differentiation of germ cells and their interaction with Sertoli and Leydig.

6.- I think that in a review of this level, it is not relevant to dedicate many comments again in relation to how the meiotic process develops in the germinal epithelium. I think that in this first part, by extending too much in describing the entire meiotic process, the main objective that the review should have is lost, which is the mechanisms that regulate these processes. In my opinion, it was not necessary to incorporate excessive details in the introduction.

7.- Finally, I believe that the authors are not specialists in these topics, since when reviewing their latest publications, their research has contributed very little to the understanding of the spermatogenic process, which is why I believe it is an important factor when writing and developing a topic that is perhaps not fully consolidated or more specialized by the authors.

So, in light of this panorama, I do not think that the review is publishable in this journal.

Reviewer 4 Report

Comments and Suggestions for Authors

The manuscript is an extensive overview of spermatogenesis, detailing the complex and highly regulated process by which spermatozoa are produced in mammals. Spermatogenesis is influenced by a multitude of factors (hormonal, genetic, and environmental factors, in particular temperature) that work in concert to ensure the production of healthy sperm. Understanding these mechanisms is crucial for addressing male infertility and developing potential therapeutic interventions.

The text is written in a very detailed manner and the references cited are numerous and pertinent. However, I would like to suggest some slight changes to make such a long manuscript easier to read.

1. First of all, the abstract is too long and detailed. The purpose of an abstract is to summarize what will be explained in detail in the manuscript. Therefore, I suggest shortening it to make the general message of the review more immediate.

2. The organization of the paragraphs in general is satisfactory. In particular, given the length of the paragraph on Gene regulation, I really appreciated Table 1 which summarizes the main points of the paragraph. Likewise, the paragraph "Cellular mechanism of Germ Cell Development" is extremely long and it would be useful to divide it into subparagraphs. Finally, it would also be useful to add a Conclusions section to help readers highlight the main points of the review.

3. The first part of the introduction does not have any references. It should cite some work (or even a textbook) from which all that information was taken.

4. According to the journal standard, References should be converted to numbers and placed in square brackets. In addition, paragraphs and subparagraphs should be numbered.

5. In Table 2, a space should be inserted between 61 and days in bovine cattle.

Author Response

  1. First of all, the abstract is too long and detailed. The purpose of an abstract is to summarize what will be explained in detail in the manuscript. Therefore, I suggest shortening it to make the general message of the review more immediate.

Agreeing with reviewer, abstract has been reduced. Lines from 10 to 31.

  1. The organization of the paragraphs in general is satisfactory. In particular, given the length of the paragraph on Gene regulation, I really appreciated Table 1 which summarizes the main points of the paragraph. Likewise, the paragraph "Cellular mechanism of Germ Cell Development" is extremely long and it would be useful to divide it into subparagraphs. Finally, it would also be useful to add a Conclusions section to help readers highlight the main points of the review.

It has been done.

  1. The first part of the introduction does not have any references. It should cite some work (or even a textbook) from which all that information was taken.

References has been included.

  1. According to the journal standard, References should be converted to numbers and placed in square brackets. In addition, paragraphs and subparagraphs should be numbered.

It has been done.

     5. In Table 2, a space should be inserted between 61 and days in bovine cattle.

It has been fixed.

Round 2

Reviewer 2 Report

Comments and Suggestions for Authors

The author has made a great improvement to the article. No further comments. Thank you.